# Phononic gravity gradiometry with Bose-Einstein condensates

Tupac Bravo[1,2], Dennis Rätzel[2], and Ivette Fuentes[3, *]

[1]*Faculty of Physics, University of Vienna, Boltzmanngasse 5, 1090 Vienna, Austria*
[2]*Institut für Physik, Humboldt-Universität zu Berlin, Newtonstrasse 15, 12489 Berlin, Germany*
[3]*School of Mathematical Sciences, University of Nottingham, University Park, Nottingham NG7 2RD, UK*
[*]*Previously known as Fuentes-Guridi and Fuentes-Schuller.*

## Abstract

Gravity gradiometry with Bose-Einstein condensates (BECs) has reached unprecedented precisions. The basis of this technique is the measurement of differential forces by interference of single-atom wave functions. In this article, we propose a gradiometry scheme where phonons, the collective oscillations of a trapped BEC's atoms are used instead. We show that our scheme could, in principle, enable high-precision measurements of gravity gradients of bodies such as the Earth or small spheres with masses down to the milligram scale. The fundamental error bound of our gravity gradiometry scheme corresponds to a differential force sensitivity in the nano-gal range per experimental realization on the length scale of the BEC.

## 1    Introduction

Since the first experimental observations of a Bose-Einstein condensation of cold atomic gases [1–3] in 1995, the possibilities to create, store and control Bose-Einstein condensates (BECs) have seen tremendous improvements. Recently, BECs were created in airplanes [4], in a free fall tower [5] and this year, on a sounding rocket in space in the course of the MAIUS 1 experiment. BECs are extreme cold (of the order of $100\,\mathrm{nK}$ and less) which makes them ideal sensor systems for a variety of sensing purposes, in particular, measurements of gravitational fields; BEC-based gravimeters and gradiometers reach extremely high sensitivities [5–9]. Furthermore, BECs are extremely small (of the order of $100\,\mu\mathrm{m}$ and less) which leads to a high potential for miniaturization, in particular, with atom chip technologies like the one presented in [7]. Such miniaturization can lead to ultra precise miniaturized gravity sensors that can be used for everyday tasks such as finding the pipework under a city [10]. The fundamental technique used for gravimetry and gradiometry with BECs is atom interferometry which was established previously with cold atomic clouds above the critical temperatures of Bose-Einstein condensation [11]. In atom interferometry, the wave function of each single atom of the BEC is split and brought into interference independently. The huge advantage of BECs for this method is their much lower temperature and size in comparison to non-condensed atomic

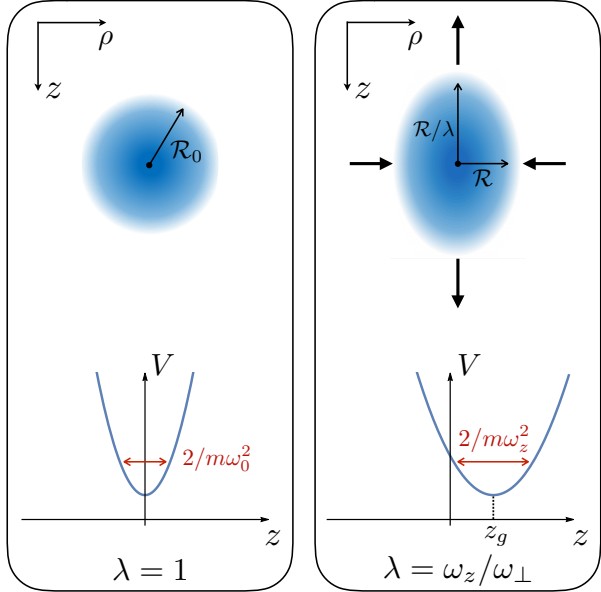

Figure 1: Deformation of the BEC due to a gravity gradient. In the absence of a gradient ($\lambda = 1$), the radio of the BEC is defined as $\mathcal{R}_0^2 = 2\mu/m\omega_0^2$. When a gravity gradient is present ($\lambda = \omega_z/\omega_\perp$), the potential is shifted to $z_g$ and the BEC takes a prolate shape with radii $\mathcal{R}$ and $\mathcal{R}/\lambda$.

gases. Collective phenomena of the atoms are not employed in state-of-the-art atom interferometry with BECs.

In contrast, the approach for gravity gradiometry presented in this article is based on phonons, collective oscillations of the atoms in a BEC. Such oscillations were the focus of experimental effort already quite early on [12–15]. Their most interesting feature is that they can be understood as quasiparticles, which was the way they were described first in [16] by Bogoliubov. Theses particles interact weakly with each other which leads to finite temperature effects like damping and a very rich phenomenology [17,18]. However, for very low temperatures and short time scales, interactions of quasiparticles in BECs are suppressed. This means that Bogoliubov quasiparticles can be used for various purposes in analogy to photons[1], in particular, quantum metrology, where quantum properties of a system are used to create sensing schemes with enhanced sensitivity in comparison to classical schemes. The use of Bogoliubov modes for quantum metrology was already theoretically investigated in [24,25]. In this article, we apply quantum metrology with Bogoliubov quasi particles to the measurement of a gravity gradient of a massive sphere as part of [26]. As a particular examples, we consider the gradiometry in the gravitational field of the Earth and that of a small massive sphere that could be hold in front of a BEC in the laboratory. The effect is very similar to the effect used to measure the thermal Casimir-Polder force which was presented in [27] and theoretically proposed in [28]; the potential of the external force adds a quadratic term to the trap potential which changes the frequencies of the collective oscillations of the BEC. In the case of the gravitational field of a

---

[1]Another purpose that is already established is the study of analogue spacetimes for Bogoliubov quasiparticles [19–21] where first experiments were performed in recent years [22,23].

massive sphere, the quadratic term in the potential is proportional to the gravity gradient. We show that the consideration of quantum metrological schemes leads to predictions of a sufficient sensitivity to measure the small effect induced by the gravity gradient of the Earth and a $20\,\mathrm{mg}$ mass on the length scale of $100\,\mu\mathrm{m}$ on an atom chip.

## 2 The external potential

Let us consider a BEC trapped in a spherically symmetric trapping potential $V_{\mathrm{trap}} = m\omega_0^2(x^2 + y^2 + z^2)/2$, where $\omega_0$ is the trapping frequency and $m$ is the mass of the atoms in the BEC. If we place the whole setup in a gravitational field represented by a Newtonian potential $\Phi$, we obtain a total potential $V = V_{\mathrm{trap}} + m\Phi$. In this article, we consider the gravitational field of a spherically symmetrical, homogeneous mass distribution which can be written as $\Phi = -MG/R$, where $M$ is the total mass, $G$ is Newton's gravitational constant and $r$ is the radial distance from the center of the mass distribution. Let us assume that the center of the mass distribution is located along the $z$-axis at a distance $R$ from the center of the trap potential. Up to second order in the spatial distances, we find $\Phi = -MG[z/R^2 - (\rho^2 - 2z^2)/2R^3]$, where $\rho^2 = x^2 + y^2$. Therefore, the total potential seen by the BEC can be written as $V = m[\omega_\perp^2 \rho^2 + \omega_z^2(z - z_g)^2]/2 + C$, where $\omega_\perp = (\omega_0^2 + MG/R^3)^{1/2}$, $\omega_z = (\omega_0^2 - 2MG/R^3)^{1/2}$, $z_g = MG/R^2\omega_z^2$ and $C = -m(MG/R^2)^2/2\omega_z^2$. We find that the linear part of the Newtonian potential leads to a shift of the equilibrium position of the BEC. By defining $z' = z - z_g$ and by neglecting the constant term $C$ (or canceling it by an additional global contribution to the time evolution of phonon modes), we obtain the potential

$$V = \frac{m}{2}(\omega_\perp^2 \rho^2 + \omega_z^2 z'^2). \tag{1}$$

Note that the trap frequencies depend on the gravity gradient, which will be the basis of its measurement in the following.

## 3 Phonon frequencies

In an axially symmetric potential such as (1), the BEC's stationary density can be approximated as $n = \mu[1 - (\rho^2 + \lambda^2 z'^2)/\mathcal{R}^2]/U_0$, where $\lambda = \omega_z/\omega_\perp$ encodes the shape of the BEC (in our case $\lambda < 1$ which corresponds to a prolate shape), $\mathcal{R}$ is the radius of the BEC in the $x$-$y$-plane, $U_0 = 4\pi\hbar^2 a_{\mathrm{scatt}}/m$, $a_{\mathrm{scatt}}$ is the scattering length and $\mu$ is the chemical potential (see [29] section 7.3.2). The radius of the BEC in the $x$-$y$-plane can be given in terms of the chemical potential as $\mathcal{R} = \sqrt{2\mu/m}/\omega_\perp$ and the total number of atoms can be derived from the density as $N_a = 8\pi\mathcal{R}^3\mu/15U_0\lambda$. This leads to the expression $\mathcal{R} = (15N_a U_0\lambda/4\pi m\omega_\perp^2)^{1/5}$.

The above approximation is called the Thomas-Fermi approximation and it describes the density profiles in experimental situations quite accurately [17] if $N_a a_{\mathrm{scatt}}/\bar{a}_{\mathrm{HO}} \gg 1$, where $N_a$ is the number of atoms in the BEC, $\bar{a}_{\mathrm{HO}} = (\hbar/m\bar\omega)^{1/2}$ and $\bar\omega = (\omega_\perp^2 \omega_z)^{1/3}$. The parabolic density profile and the spheroidal shape of the BEC give rise to a specific set of modes of density perturbations that was presented in [30]: the density perturbations have the spatial dependence $\delta n \propto r^l Y_{lm}(\theta, \phi)$ in spherical coordinates $r = \sqrt{\rho^2 + z^2}$, $\theta = \arccos(z/r)$ and $\phi = \arctan(y/x)$, where $Y_{lm}(\theta, \phi)$ is the spherical harmonic function with angular momentum $l$ and the $z$-component of the angular momentum vector $m = \pm l$ or $m = \pm(l - 1)$. We denote these two types of modes as $\delta n_{l,l}$ and $\delta n_{l,l-1}$. The corresponding angular frequencies are given as $\omega_{l,l}^2 = l\omega_\perp^2$ and $\omega_{l,l-1}^2 = (l-1)\omega_\perp^2 + \omega_z^2$, respectively. We can expect the effect of the gravity gradient to be much smaller than that of the trap potential. Therefore, we can approximate the frequencies as $\omega_{l,l} \approx \sqrt{l}\left[\omega_0 + MG/(2R^3\omega_0)\right]$ and

$\omega_{l,l-1} \approx \sqrt{l}\,[\omega_0 + (l-3)MG/(2lR^3\omega_0)]$. The modes $\delta n_{l,l}$ and $\delta n_{l,l-1}$ contain information about the gravity gradient $\epsilon_{\mathrm{grad}} := 2MG/R^3$ as an external parameter. Therefore, it is possible to estimate its value from a measurement on these modes.

## 4 Single mode sensitivity

A lower bound for the relative error $\delta_\epsilon$ of any estimation of a given parameter $\epsilon$ imprinted on a given mode can be obtained from the quantum Cramér-Rao bound (QCRB). The QCRB can be expressed in terms of the quantum Fisher information (QFI) $H_\epsilon$ and the number of measurement repetitions $N_{\mathrm{rep}}$ as $\Delta_\epsilon = 1/\sqrt{N_{\mathrm{rep}}H_\epsilon}$. The corresponding relative error bound is obtained as $\delta_\epsilon = 1/|\epsilon|\sqrt{N_{\mathrm{rep}}H_\epsilon}$. Since $\epsilon_{\mathrm{grad}}$ is imprinted only on the frequency, we obtain the minimal relative error for the estimation of $\epsilon_{\mathrm{grad}}$ by Gaussian error propagation from the minimal error of an estimation of a phase change $\Delta\phi = \Delta\omega\,t$ as $\delta_\epsilon = \delta_{\Delta\phi}/|d\ln(\Delta\phi)/d\ln\epsilon_{\mathrm{grad}}|$, where $\Delta\omega = \sqrt{l}\,\epsilon_{\mathrm{grad}}/4\,\omega_0$ for the mode $\delta n_{l,l}$ and $\Delta\omega = (l-3)\,\epsilon_{\mathrm{grad}}/4\,\omega_0\sqrt{l}$ for the mode $\delta n_{l,l-1}$. Therefore, we find that $\delta_\epsilon = \delta_{\Delta\phi}$. If we assume that the initial state of the mode under consideration is a Gaussian state and we are only measuring one single mode, the optimal sensitivity for the measurement of a phase change is reached for a squeezed vacuum state and the corresponding QFI becomes $H_{\Delta\phi} = 8N_r(N_r + 1)$ [31,32], where $N_r = \sinh^2 r$ is the number of squeezed phonons, where $r$ is the squeezing parameter [2]. We obtain

$$\delta_{\epsilon_{\mathrm{grad}}} = \frac{2\omega_0^2}{\alpha(l)\epsilon_{\mathrm{grad}}}\frac{1}{\sqrt{l}\,\omega_0 t\sqrt{2N_{\mathrm{rep}}N_r(N_r + 1)}}\,, \tag{2}$$

where $\alpha(l) = 1$ for the modes $\delta n_{l,l}$ and $\alpha(l) = |(l-3)/l|$ for the modes $\delta n_{l,l-1}$ and where Heisenberg scaling is reached when $N_r \gg 1$ [31].

## 5 Measurement sensitivity

Let us evaluate the relative error bound in Eq. (2) for two specific situations; the gravitational field of the Earth and the gravitational field of a sphere of tungsten or gold. Let us assume that the trapping frequency is $\omega_0 = 2\pi \times 0.2\,\mathrm{Hz}$. For a rubidium-87 BEC of $10^6$ atoms this would lead to a radius of $120\,\mu\mathrm{m}$ and a central density of $n(0) \sim 10^{11}\,\mathrm{cm}^{-3}$, which is fully in the currently experimentally accessible regime. An optimistic estimate for future technology would be an assumption of $10^8$ rubidium atoms. This would lead to a radius of $\mathcal{R} = 300\,\mu\mathrm{m}$ and a central density of $n(0) \sim 10^{12}\,\mathrm{cm}^{-3}$. The small density of the BEC is an advantage as it leads to a long half life time of the BEC density. In Section 5.4 of [29] and in [33, 34] it was shown that the density depends on time as $dn(t)/dt = -Dn(t)^3$, where $D$ is the decay constant. Therefore, after solving the differential equation, we find a quadratic dependence of the density half life time on the inverse density, i.e. $t_{\mathrm{hl}} = 3/(2Dn(0)^2)$. In [35], an experiment with rubidium atoms was presented where the corresponding decay constant was found to be $D = 5.8 \times 10^{-30}\,\mathrm{cm}^6\,\mathrm{s}^{-1}$. For a density of the order of $10^{12}\,\mathrm{cm}^{-3}$, this leads to a theoretical half life time of the order of the order of $10^5$ s. The second limiting time scale that one has to consider is the inverse damping rate of the phonons. For BECs of temperatures $T$ below or of the order of the chemical potential divided by the Boltzmann constant $k_B$, the damping rate $\gamma$ of low frequency phonon modes was given in [36]. We find $\gamma \sim \sqrt{l}\omega_0(k_BT/\mu)^{3/2}(n(0)a_{\mathrm{scatt}}^3)^{1/2}$. Assuming $l = 3$ and a temperature of the BEC of 0.1 nK, which can be achieved in experiments [5, 37], we find inverse damping rates of the order of $10^3$ s and $10^4$ s

---

[2]In [32], the squeezing parameters is defined so that the squeezing operator becomes $S = \exp\left(-r(e^{i\chi}\hat{a}^{\dagger 2} - e^{-i\chi}\hat{a}^2)/2\right)$

for $10^6$ and $10^8$ rubidium atoms, respectively. Therefore, we assume a duration of 100 s for each experiment.

For the number of independent consecutive measurements, we assume a value of $10^4$ which corresponds to a total measurement time of one and a half weeks. We set $N_r = 10^3$ and $N_r = 10^4$ for the number of squeezed phonons in the cases of $N_a = 10^6$ and $N_a = 10^8$ atoms, respectively (which corresponds to a squeezing parameter of $r \sim 4$ and $r \sim 5$, respectively). We use the mode number $l = 3$. For the case of the gravitational field of the Earth and $R$ of the same order as the radius of the Earth, we find that $\omega_0^2 R^3 / 2MG$ is of the order of $10^6$. This leads to a relative error bound of the order of $10^{-2}$ and $10^{-3}$ for $10^6$ and $10^8$ atoms, respectively. Hence in principle, it is possible to measure the gravitational gradient due to the gravitational field of the Earth on the length scale of a BEC using the phonons in the condensate.

A 20 mg gold or tungsten sphere has a radius of 0.63 mm. Assuming a distance between the center of the BEC and the center of the sphere of 1 mm, and with $\omega_0 = 2\pi \times 0.2$ Hz, we find that $\omega_0^2 R^3 / 2MG$ is again of the order $10^6$. We thus find a relative error bound of the order of $10^{-2}$ and $10^{-3}$ for $10^6$ and $10^8$ atoms, respectively, by using the same system parameters as above. The reason for this scaling is that by considering $R$ to be always of the same order as the radius of the sphere independently of its mass, we obtain that $2MG/R^3 \sim 8\rho_M G$, where $\rho_M$ is the mass density of the sphere.

The absolute sensitivity of the scheme is given as $\epsilon_{\text{grad}} 10^{-2} \sim 10^{-7}\,\text{s}^{-2}$ and $\epsilon_{\text{grad}} 10^{-3} \sim 10^{-8}\,\text{s}^{-2}$ for $10^6$ and $10^8$ atoms in the BEC, respectively. On the length scale of the BECs, such small gradients correspond to gravitational force differences of $10^{-9}$ gal ($10^{-12}\,g$) for $10^6$ atoms and $10^{-10}$ gal ($10^{-13}\,g$) for $10^8$ atoms, where we considered $10^4$ repetitions of the experiments. The single shot sensitivity would be comparable to a differential force sensitivity of the order of $10^{-7}$ gal ($10^{-10}\,g$) and $10^{-8}$ gal ($10^{-11}\,g$), respectively.

# 6 Interferometric estimation

To have a chance to reach the relative error bound given in Eq. (2), there has to be a phase reference to compare the phase change of the probe state with. In particular, the frequency of the reference has to be exactly $\sqrt{l}\,\omega_0$. One possibility to avoid this would be to use two modes of perturbations of the BEC density and compare them with each other. Let us consider two modes with the same angular momentum $l$ but different quantum number $m$. The difference of the two frequencies can be approximated as

$$\Delta\omega_l = \omega_{l,l} - \omega_{l,l-1} \approx \frac{3\epsilon_{\text{grad}}}{4\sqrt{l}\,\omega_0}\,. \tag{3}$$

The frequency difference would lead to an accumulated phase difference between the modes $\Delta\phi_l = \Delta\omega_l t$. Note that the frequency difference (3) decreases with increasing angular momentum $l$. This is in contrast to the change of frequency of each single mode due to the gravitational field that we considered above.

Although, so far, there has not been an experimental realization of an interferometer scheme for phonons in a BEC, the possibility to deform BECs quite strongly and to control the interactions in the BEC with Feshbach resonances make it feasible that controlled, strong coupling of phonons may be implemented in the future. This would enable multi-mode operations that can be used for the implementation of different interferometric schemes. In metrology with optical modes, there are two distinct classes of interferometric optical devices; SU(2) and SU(1,1). In what follows, we will use the term "two-mode two-beamsplitter interferometer" (TTI) to refer to setups that perform

one beamsplitting operation, a phase and another beamspitting operation like the Mach-Zehnder interferometer in optics. In contrast to the beamsplitting in an optical Mach-Zehnder interferometer, the abstract notion of beamsplitting does not indicate a spatial separation between the modes. In our case, the two modes used are separated by frequency. SU(2) schemes only consider passive optical elements and the SU(1,1) schemes contain active optical elements. In all schemes the optimal QFI depends on the total number of particles in the two modes of the TTI which we denote as $\bar{N}$ in the following. In [38], it is shown that the optimal QFI for a phase measurement for the SU(2) scheme is reached for two identically squeezed and coherently displaced states at the two input ports of the first beam splitter of the TTI. The total number of squeezed particles has to be $2/3$ of the total number of particles $\bar{N}$. For the case of $\bar{N} \gg 1$, the optimum becomes $H_{\Delta\phi}^{\mathrm{SU(2)}} \approx 8\bar{N}(\bar{N}+2)/3$. In contrast to the SU(2) scheme, the SU(1,1) scheme uses active elements instead of passive beam splitters. In optics, parametric amplifiers (OPA) are used, where light modes interact non-linearly and they are additionally squeezed. Also in [38], it is shown that the optimal QFI for the SU(1,1) scheme is given as $H_{\Delta\phi}^{\mathrm{SU(1,1)}} \approx 4\bar{N}(\bar{N}+2)/3$ for $\bar{N} \gg 1$. To reach this QFI, two coherent states with the same number of particles have to be injected into the two ports of the TTI and the number of squeezed particles that are created at the active elements should be $2/3$ of the total number of particles in the modes of the TTI. If the number of squeezed particles is fixed, $H_{\Delta\phi}^{\mathrm{SU(2)}}$ and $H_{\Delta\phi}^{\mathrm{SU(1,1)}}$ differ from the optimal single mode QFI that we discussed above approximately by a factor $3/4$ and $3/8$, respectively.

In [39], a particular adaptation of the SU(1,1) scheme is presented that is also beneficial when the initial number of squeezed particles is much smaller than the total number of particles. It is called "pumped-up" SU(1,1) interferometry by the authors of [39] because the active optical elements are additionally pumped by a strong coherent field with particle number $N_\alpha = |\alpha|^2$ and coherent parameter amplitude $|\alpha|$. It is shown that the optimal QFI for the pumped-up SU(1,1)-scheme becomes $H_{\Delta\phi}^{\mathrm{pu}} = N_\alpha\, e^{2r}/4$. For $N_r = \sinh^2 r$ and $N_r \gg 1$, we find that $H_{\Delta\phi}^{\mathrm{pu}}$ differs from the one mode QFI by a factor $1/16$. However, in situations where the number of coherent particles is much larger than the number of squeezed particles, the application of the pumped up SU(1,1) interferometry can be truly beneficial. A further discussion with the possibility of its inclusion for the detection of gravitational waves with BECs is presented in [40]. The incorporation of these elements and TTIs into our proposal will be left for future work.

Eventually, we find that by using advanced interferometric schemes, the sensitivity of our phononic measurement scheme for gravity gradiometry can be improved by one order of magnitude. Therefore, for the cases we consider here, this leads to a relative error bound of the order of $10^{-3}$ and $10^{-4}$ for $10^6$ and $10^8$ atoms, respectively, and an absolute error bound $\epsilon_{\mathrm{grad}}10^{-3} \sim 10^{-8}\,\mathrm{s}^{-2}$ and $\epsilon_{\mathrm{grad}}10^{-4} \sim 10^{-9}\,\mathrm{s}^{-2}$ for $10^6$ and $10^8$ atoms in the BEC, respectively, and $10^4$ measurement repetitions.

# 7  Comparison

In the following, we want to compare the sensitivity of our measurement scheme with the state of the art in gravity gradiometry with cold atoms, which is single particle matter wave interferometry with free falling atoms as presented in [9]. In order to obtain a fair comparison, we consider the experimental setup presented in [9] scaled to the length scale of $500\,\mu\mathrm{m}$, the characteristic length of our setup for $N_a = 10^8$. Furthermore, we give the fundamental limit of sensitivity for this setup using the same parameters as we used for our setup which are the atom number $N_a = 10^8$ and the number of squeezed particles $N_r = 10^4$. The QFI for phase measurements by single particle matter wave interferometry is given as $H_{\Delta\phi} \approx 8N_r(N_a + 2N_r)$, where a single mode squeezed coherent

state with $N_r \gg 1$ is assumed as the probe state. In the case of gravity gradiometry, the phase is the tidal phase $\phi_{\text{tidal}} = \hbar n^2 k^2 \epsilon_{\text{grad}} t^3/(2m)$, where $k = 2\pi/\lambda$ and $\lambda = 1.56\,\mu\text{m}$ is the laser wave length [41]. Restricting the size $s$ of the entire setup limits the free fall time $t_{\text{free}} \leq 2\sqrt{2s/g}$, where $g$ is earth's gravitational acceleration, and the momentum kick splitting the atoms' wave functions $n \leq ms/(\hbar k\, t_{\text{free}})$, which leads to $\phi_{\text{tidal}} \leq m\epsilon_{\text{grad}} s^{5/2}/(\sqrt{2g}\,\hbar)$. For a characteristic length scale of $600\,\mu\text{m}$, we obtain the maximal (and optimal) value $n = 18$ (an even number as Bragg transitions are employed for momentum transfer) and $t_{\text{free}} = 20\,\text{ms}$. Based on these parameters and the gravity gradient in the earth's gravitational field close to its surface, we find a relative error bound $\delta_{\epsilon_{\text{grad}}}$ of the order of $10^{-4}$ for $10^4$ repetitions, which is of the same order of magnitude as the sensitivity we have predicted for our measurement scheme employing advanced interferometric schemes. The technology of trapped atom interferometry currently under development offers increased integration time with respect to the state of the art in matter wave interferometry described above [42]. As in our approach, the atoms are trapped in potential wells, can be brought close to gravitational sources and held there for an increased time. Let us assume that two trapped atom interferometry experiments are performed at two different points in a gravitational field with gradient $\epsilon_{\text{grad}}$ at a distance of $L \sim 600\,\mu\text{m}$, the order of magnitude of the BEC length in our proposal. To measure the differential acceleration between these points, the split of the atoms' wave functions $\Delta z$ at the two points has to be much smaller than the distance between the points. The difference between the two gravitationally induced phase differences for the two interferometers is $\Delta\phi = m\,L\,\epsilon_{\text{grad}}\,\Delta z\, t/\hbar$. Accordingly, the relative sensitivity of this setup for measurements of $\epsilon_{\text{grad}}$ becomes $\delta_{\epsilon_{\text{grad}}} = \delta_{\Delta\phi} = \Delta_\phi/\Delta\phi = (\sqrt{N_{\text{rep}} H_{\Delta\phi}} \Delta\phi)^{-1}$. Considering an integration time of $t \sim 100\,s$, a split of each interferometer of $\Delta z \sim 30\,\mu\text{m}$, the mass of Rubidium atoms and using the parameters $N_a = 10^8$ and $N_r = 10^4$ again, we obtain the fundamental relative error bound $\delta_{\epsilon_{\text{grad}}} \sim 10^{-6}$ for $10^4$ repetitions for estimations of the gravity gradient in the earth's gravitational field close to its surface (of the order of $10^{-6}\text{s}^{-2}$). Independently, this result shows the great potential of using trapped atoms instead of free falling atoms for gradiometry, and establishes a fundamental error of two orders of magnitude better than the fundamental error bound of our measurement scheme.

# 8   Conclusions

Our considerations show that it should be possible in principle to use collective oscillations of a BEC to measure the gravity gradient of the Earth or a small 20 mg sphere of tungsten. The usual approach to gravity gradiometry with BECs is the measurement of a differential force with atom interferometry which is done on length scales of centimeters to meters. In our approach, the gravity gradient is measured on the length scale of the BEC which is of the order of $100\,\mu\text{m}$.

We found relative error bounds of the order of $10^{-3}$ and $10^{-4}$ for $10^6$ and $10^8$ atoms, respectively, and $10^4$ repetitions of the experiment for the measurement of the gravity gradients in the setup mentioned above. If the number of atoms in a BEC cloud can be further increased to $10^{10}$ in years to come, we could reach a relative error bound of $10^{-5}$. Our comparison to the fundamental sensitivity limit of state of the art gradiometry with free falling matter wave interferometry showed similar sensitivity. The technology of trapped atom interferometry currently under development promises a fundamental limit that is two orders of magnitude better, which shows the great potential of gradiometry with trapped atoms. The sensitivity limit of our scheme may be improved by using other modes than the pure angular momentum modes we considered in this work. However, no analytical expressions for the energies of such modes seem to be known [30]. Therefore, numerical methods would have to be applied.

From a measurement of the gravity gradient of a sphere of known mass one can infer the grav-

itational constant when the distance to the center of the sphere is also known. If the density of the sphere is sufficiently homogeneous and the mass and the distance to the center are known with high enough precision, the gravitational constant could be inferred with a relative error of up to $10^{-3}$, which is only one order smaller than what was achieved in gravimetry experiments with atom interferometers [43]. Here, the use of small masses of the order of $10\,\mathrm{mg}$, like the one used in the present work, is of course beneficial as their homogeneity can be ensured and their mass can be measured more precisely. Additionally, a periodic movement of the sphere in front of a trapped BEC can lead to further effects such as resonance, that can, in principle, be measured as is studied in [44].

One interesting aspect about gradiometry is that it does not rely on any inertial effect in contrast to gravimetry. Therefore, gradiometry with BECs could be performed in free fall [4,5] and in space. Furthermore, the measurement of the gravitational gradient corresponds to the measurement of particular components of the curvature tensor of spacetime in the Newtonian limit. Curvature is the only local physical effect in general relativity which makes gradiometry the only true metrology of the gravitational field in that context. Note that our gradiometry scheme would measure the curvature on a length scale of the wave function of the quantum system used for the sensing process similar to the atom interferometric experiment presented in [9]. However, before any such claims can be made in full generality, the gradiometry scheme has to be described fully relativistically.

Some technical questions have to be solved before our proposed gradiometry scheme can be implemented. The basic ingredients are the possibility to create squeezed states of Bogoliubov quasiparticles in BECs, to let several quasiparticle modes interact strongly in a controlled way and to read them out with very high precision. Squeezing is well established for photons in quantum optics [45,46] and for phonons in quantum optomechanics [47,48]. For Bogoliubov quasiparticles, squeezing could be achieved by periodically changing the shape or the atom-atom interaction strength of the BEC as it was done in [49] to create correlated pairs of quasiparticles. The controlled coupling of different quasiparticle modes seems to be the technical part that needs the most development. This would amount to the implementation of a "beam splitter" for phonon modes. Although the implementation of such coupling lies beyond our awareness, it is conceivable, in principle, that phonon-phonon interactions could be employed for an implementation via manipulation of the BEC's shape. The read out of the modes may then be established with a technique sometimes denoted as heterodyne detection [50]. Finally, the introduction of optomechanics into BEC-based setups such as [51–53] may lead to exciting proposals from which the precision of certain measurements might benefit.

# Acknowledgments

The authors thank Richard Howl and David E. Bruschi for many useful remarks and discussions. TB acknowledges funding by CONACYT under project code 261699/359033. DR acknowledges financial support by the Humboldt Foundation through the Feodor Lynen Return Fellowship and by the European Commission through the MSCA Individual Fellowship PhoQuS-G. IF acknowledges financial support by the EPSRC award: INSPIRE Physical Sciences: Levitation-based quantum gravimeter Ref:EP/M003019/1 and the Penrose Institute.

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
