# Peer review of "Phononic gravity gradiometry with Bose-Einstein condensates"

_SciPost Physics_

## Round 2 · Referee Report · Anonymous (Referee 1) · 2020-6-15

Strengths

The idea of using collective modes in BECs looks like an interesting way to overcome the difficulty imposed by atom-atom interactions in interferometric schemes, while still taking advantage of the low temperature and the large number of particles of condensates. The quantitative values displayed by the authors seem optimistic about the long-time promise of such schemes.

Weaknesses

The authors' estimates for the sensitivity of the proposed device are based on a string of assumptions about the possibility of realizing strongly squeezed states of collective excitations in BECs. In my opinion, learning how to generate and manipulate quantum states of collective excitations constitutes an entire (and very exciting) new field of research on its own, which will require substantial efforts by the wide community in the coming years before getting to be useful for precision measurements. In this sense, I find the manuscript to be slightly overoptimistic and the authors do not present any specific and concrete proposal to generate and manipulate squeezing of Bogoliubov modes.
Along related lines, they do not discuss possible spurious effects due to experimental fluctuations and nonlinear corrections beyond Bogoliubov that might have a dramatic impact on the precision of the measurement at high squeezing levels.

Report

The manuscript by Bravo et al. reports a theoretical study of collective modes in atomic Bose-Einstein condensates as a probe for gravity gradients. The idea is to measure the differential frequency shift between pair of modes that are differently affected by gravity. Under quite strong assumptions, the manuscript reports that such a configuration could actually improve the performance of detectors in a significant way. On one hand, I find that the idea of using collective excitations rather than single particles as a probe for precision measurements is a quite intriguing one and could allow to fully unchain the potential of BECs (in particle number and temperature) for interferometry. On the other hand, I am afraid that the theoretical analysis carried out in the manuscript is not complete nor detailed enough for publication on a top-notch journal like SciPost. The calculations reported in the manuscript are a sort of straightforward estimate of the error based on some assumptions on the atom sample and on the measurement procedure. In particular, I understand that the efficiency of the proposal crucially depends on the ability to squeeze the quantum state of Bogoliubov modes to quite important values. Unfortunately, this is not something that -to the best of my knowledge- has ever been demonstrated in the lab nor has been addressed by convincing theoretical proposals. The fact that only a few generic statements are devoted to this crucial issue in the conclusion section and no concrete proposal is put forward to solve it strongly reduces the scientific strength of the manuscript. Another dangerous issue that jumps to my mind is the level of run-to-run stability that the BEC must have to perform the proposed experiment: for instance, how are the authors dealing with the usually significant shot-to-shot fluctuations in the total atom number? this was not too much a concern for the dipole oscillation mode used in ref.27, but may be a source of strong shot-to-shot fluctuations for the higher-l modes considered here.

In what follows, I give a list of technical remarks that might turn out useful in view of resubmission to some other, more specialized journal:

  • *Most important* According to the authors' estimates, an initial number N_r~10^3 (or 10^4) or squeezed phonons is needed to optimize the measurement. While this may seem a small number compared to the total atom number of N_a~10^6 (or 10^8), the authors did not estimate the actual value of non-condensed atoms that exist in such a state. Since they are working with collective modes, I suspect that a quite large Bogoliubov coefficient may enter into the conversion from the quasi-particle basis to the atom basis. Since the condition for the validity of the Bogoliubov approximation is typically expressed in terms of the non-condensed fraction, this may impose non-trivial additional limitations to the proposal. Furthermore, along similar lines, the authors should also justify in a quantitative way their assumption that such a large squeezing does not introduce any nonlinear frequency shift of the Bogoliubov modes. With no quantitative analysis, I am not sure that such an effect does not have a devastating effect on their proposal.

-on pag.2, the sentence "Theses particles ... like damping... phenomenology." is to my opinion not fully correct and should be amended.

-on pag.4, perhaps it is just by my ignorance, but I do not understand why d\ln(Delta\phi)/d\ln \epsilon_{grad} (and not a standard derivative with no \ln's) enters in the formula for the error propagation.

-on pag.5, the authors speak about "a relative error bound" but I do not understand how this quantity is defined and to what it is to be compared to assess the performance of the experiment.

-again, a few lines later, they speak about "absolute sensitivity" without giving a definition for it. Non-expert readers would enjoy seeing it definied explicitly.

-on pag.7, in the middle of Sec.7 the authors start speaking about "two trapped atom interferometry experiments". They should explain more clearly the need for such a more complex configuration and explain how this connects with the rest of their analysis.

-the authors never mention the optomechanics experiments performed by the Esslinger group, e.g. https://science.sciencemag.org/content/322/5899/235.abstract Since they quote quantum optomechanics as a strategy to squeeze phonons, I am curious to know if such set-ups could be of any help for their proposal.

-the authors briefly mention analog gravity but do not discuss how such a framework has already led to some experimental evidence of two-mode squeezing effects and of the consequent entanglement, https://www.nature.com/articles/nphys3863 Could such a scheme be useful for their proposal? Or do they share the concerns about the validity of this work put forward by some researchers and intentionally chose not to cite it?

---

## Editorial Decision

awaiting_resubmission